# Maternal Plasma miRNAs as Early Biomarkers of Moderate-to-Late-Preterm Birth

**DOI:** 10.3390/ijms25179536

**Published:** 2024-09-02

**Authors:** Farha Ramzan, Jing Rong, Claire T. Roberts, Justin M. O’Sullivan, Jo K. Perry, Rennae Taylor, Lesley McCowan, Mark H. Vickers

**Affiliations:** 1Liggins Institute, University of Auckland, Auckland 1142, New Zealand; f.ramzan@auckland.ac.nz (F.R.); jing.rong@auckland.ac.nz (J.R.); justin.osullivan@auckland.ac.nz (J.M.O.); j.perry@auckland.ac.nz (J.K.P.); 2Flinders Health and Medical Research Institute, College of Medicine and Public Health, Flinders University, Adelaide 5001, Australia; claire.roberts@flinders.edu.au; 3Maurice Wilkins Centre, University of Auckland, Auckland 1142, New Zealand; 4Department of Obstetrics and Gynaecology, Faculty of Medical and Health Science, University of Auckland, Auckland 1142, New Zealand; r.taylor@auckland.ac.nz (R.T.); l.mccowan@auckland.ac.nz (L.M.)

**Keywords:** preterm birth, miRNA, biomarker, pregnancy, epigenetics

## Abstract

Globally, preterm birth (PTB) is a primary cause of mortality and morbidity in infants, with PTB rates increasing worldwide over the last two decades. Biomarkers for accurate early prediction of PTB before the clinical event do not currently exist. Given their roles in the development and progression of many disease states, there has been increasing interest in the utility of microRNAs (miRNAs) as early biomarkers for pregnancy-related disorders, including PTB. The present study was designed to examine potential differences in miRNA abundances in maternal plasma from mothers with infants born following a moderate to late (28–36 weeks’ gestation, n = 54) spontaneous PTB (SPTB) compared to mothers with matched term infants (n = 54). Maternal plasma collected at 15 weeks’ gestation were utilised from the Auckland and Adelaide cohorts from the Screening for Pregnancy Endpoints (SCOPE) study. miRNAs in plasma were quantified using the NanoString nCounter expression panel (800 miRNAs). The top four most abundant miRNAs were significantly decreased in the plasma of mothers in the SPTB group with results consistent across both cohorts and pathway analysis was undertaken to examine the biological processes linked to the dysregulated miRNAs. The top candidate miRNAs (miRs-451a, −223-3p, let-7a-5p, and -126-3p) were linked to gene pathways associated with inflammation, apoptosis, and mitochondrial biogenesis. Moreover, miRNAs were consistently less abundant in the plasma of mothers of preterm infants across both sites, suggesting potential global dysregulation in miRNA biogenesis. This was supported by a significant downregulation in expression of key genes that are involved in miRNA biogenesis (*DROSHA*, *DICER*, and *AGO2*) across both sites in the SPTB group. In summary, the present study has identified miRNAs in maternal plasma that may provide predictive utility as early biomarkers for the risk of later SPTB. Importantly, these observations were conserved across two independent cohorts. Further, our data provide evidence for a persistent decrease in miRNA abundance in mothers who later experienced an SPTB, which is likely to have widespread consequences for gene regulation and epigenetic processes.

## 1. Introduction

At a global level, more than 1 in 10 babies are born prematurely (before 37 weeks of gestation), equating to more than 15 million preterm births that result in over one million newborn deaths [1]. Preterm birth (PTB) and its associated complications have been reported to account for over 30% of the 3.1 million neonatal deaths per year globally. Following pneumonia, PTB is the second most common cause of death in children under 5 [1,2,3]. PTB also results in an increased risk of death arising from other causes, particularly that of neonatal infections [4,5], with PTB being the leading factor contributing to child deaths across almost all low- and middle-income countries [2].

PTBs can be divided into spontaneous (SPTB) or iatrogenic PTB. SPTB refers to an unintentional, unplanned delivery before the 37th week of pregnancy and can result from a number of reasons, such as infection or inflammation, although the causal factors for most SPTBs usually remain unknown. Iatrogenic PTB is a planned delivery that occurs before 37 weeks of pregnancy due to maternal and/or foetal causes. However, in some cases, such deliveries also occur with no apparent medical indication. Of all PTBs, approximately 60% occur after the spontaneous onset of labour. In approximately 40% of PTBs, pregnancy complications result in babies being delivered early. Although a previous SPTB places women at an elevated risk for a recurrence, a majority present in women without prior history, particularly in those who are in their first pregnancy. Of note, a recent overview of systematic reviews by Salmeri et al. highlighted that singletons conceived through in vitro fertilisation (IVF) or intracytoplasmic sperm injection have, on average, a 2-fold increased risk of SPTB or iatrogenic PTB compared to those conceived naturally [6]. As such, early biomarkers that were accurate predictors for the later risk of SPTB, prior to the clinical event, would allow for improved care and the potential for targeting existing and novel therapeutic approaches to mitigate PTB, which could improve outcomes for both mothers and infants.

MicroRNAs (miRNAs) represent a class of small noncoding RNAs of 18–24 nucleotides in length that regulate gene expression and a range of cellular functions by inhibiting translation or directing degradation of messenger RNA. Activities of miRNAs, therefore, represent key epigenetic mechanisms underlying gene regulation that can mediate complex processes such as cell growth and differentiation, tissue remodelling, and stress responsiveness that, under certain conditions, can play a key role in many disease conditions [7], including complications related to pregnancy [8,9]. Given their stability in circulation, miRNAs have gained increasing attention as potential biomarkers and therapeutic targets in the setting of a range of complications related to pregnancy. However, whether differential patterns of maternal circulatory miRNAs possess utility as potential early biomarkers for later pregnancy outcomes remained to be fully investigated. There remains a lack of data in this rapidly emerging research area, but recent work has highlighted the potential for miRNAs as biomarkers for pregnancy complications, including pre-eclampsia and babies born small for gestational age [10,11,12,13]. A recent systematic review also highlighted a number of miRNAs in the first trimester that were associated with a range of pregnancy complications, including gestational hypertension, pre-eclampsia, and PTB [9].

There has been increasing interest in the potential utility of miRNAs in the setting of PTB, although the data remain inconsistent. Differences in cervical miRNA profiles have been reported in women later destined to have an SPTB compared to those that delivered at term [14]. A further study also investigated miRNAs in blood from women at risk for PTB but reported no significant differences in those women who experienced a PTB versus those women who delivered at term [15]. However, maternal samples were profiled at the time of assessment or admission for PTB and, therefore, have limited “predictive” utility. In addition, a recent study examining the expression profiles of miRNAs in placental tissue in the setting of preterm prelabour rupture of membranes and subsequent SPTB highlighted differences in profiles between term and SPTB groups but did not provide a predictive measure or any clinical utility as the analysis was conducted at the time that pregnancy complications were already manifest [16]. Winger et al. reported differences in miRNA profiles in maternal peripheral blood cells during the first trimester of pregnancy and reported predictive utility in identifying later SPTB. Further work by Cook et al. highlighted first-trimester circulating miRNAs that were predictive of subsequent cervical shortening and PTB although the PTB cohort size was relatively small [17]. More recent work by Illarionov et al. has also detailed distinct miRNA profiles in maternal plasma in women at an elevated risk for PTB, although, again, the cohort size was small [18]. Some of the discrepancies in the reports to date likely link, in some cases at least, to low statistical power as well as the range of methodological platforms used. Further, in addition to different sampling time points and the known temporal changes in miRNAs across pregnancy [19], some of the variation in results to date could, in part, also reflect cohort demographics, with ethnicity suggested to contribute to inconsistent validation of predictive miRNA targets and clinical outcomes [20,21].

A further potential limitation of some studies to date is not differentiating between iatrogenic PTB and SPTB. Recent work has highlighted that iatrogenic PTB and SPTB represent quite different entities with unique pathologic features and, therefore, quite different miRNA-related pathways are likely to be involved across the two pregnancy complications [22]. In the setting of SPTB, we have published preliminary evidence to show that circulating miRNAs and associated gene products have the potential to provide viable targets for clinical biomarkers for risk of SPTB in maternal plasma from mothers in mid-pregnancy (20 weeks’ gestation) [23]. The current study therefore aimed to expand on these original findings and investigate whether miRNAs in plasma from mothers during early-mid pregnancy (15 weeks) have utility as later predictors of moderate to late SPTB. Further, this study spanned two cohorts to examine whether findings could be replicated across independent cohorts where standardised collection protocols had been utilised.

## 2. Results

Demographics of these two cohorts are presented in Table 1. There were no differences in baseline characteristics between Cases and Controls across the two maternal groups other than gestational age at time of delivery. Overall, at a cohort level, maternal age and socioeconomic index scores were lower in the Adelaide cohort as compared to that of the Auckland cohort.

There was an overall downregulation in miRNAs in the plasma of mothers at 15 weeks’ gestation who later experienced an SPTB as evidenced in the heat maps for both Auckland and Adelaide cohorts (Figure 1a,b). Of the top 12 most abundantly expressed miRNAs, all were significantly downregulated (*p* < 0.05, Figure 2) in the SPTB group as compared to term controls with the exception of let-7a-5p in the Auckland cohort, which trended towards a decrease (*p* = 0.09). Of note, the top 12 most differentially expressed miRNAs were tightly conserved across the Auckland and Adelaide cohorts (Table 2), with similar patterns of downregulation in the SPTB group observed across both sites. The most highly differentially expressed miRNAs conserved across both sites were miR-451a, 223-3p, let-7a-5p, and -126-3p. 

ROC curves generated for miR-451a demonstrated a good predictive fit for later SPTB across both Auckland and Adelaide cohorts, with area under the curve (AUC) values of 0.804 and 0.833, respectively (Figure 3). miR-223-3p provided a high predictive value for SPTB with an AUC of 0.867 in the Adelaide cohort but this goodness of fit was reduced in the Auckland cohort (AUC = 0.673) (Appendix A). Similarly, while let-7a-5p and mIR-126-3p provide a good predictive fit for SPTB in the Adelaide cohort with AUCs of 0.781 and 0.787, respectively, these values were lower and of less predictive utility in the Auckland cohort with 0.652 and 0.688, respectively.

### 2.1. Gene Abundance

Abundance of *Dicer*, *Drosha*, *Ago1*, and *Ago 2* was significantly reduced in the SPTB group as compared to Controls in both the Auckland and Adelaide SCOPE cohorts (Figure 4a–h). 

### 2.2. miRNA Pathway Analysis

The miRNA–gene network and functional analysis were performed for the top 14 commonly differentiated miRNAs across both cohorts. A total of 2975 genes (both strong and weak interactions) were shown to be targeted by these miRNAs (Appendix A). To ascertain the strongest miRNA–gene interaction, we used the betweenness filter cut-off of 2.0. Based on this analysis, the miRNAs were shown to map to 240 genes (Figure 5). Based on our functional enrichment analysis of these 240 targets, a total of 45 pathways (FDR ≤ 0.05) were significantly enriched by these genes using KEGG. Among these, pathways that are thought to have important roles in the pathogenesis of SPTB were prioritised for further analysis. According to these results, 16 pathways were demonstrated to be enriched with genes involved in immune and inflammatory-related processes, such as cell cycle, focal adhesion, Jak-STAT signalling, ErbB signalling, chemokine signalling, and neurotrophin signalling pathways (Table 3). Moreover, our GO term analysis showed significant enrichment of the target genes in 100 GO-BP terms (Appendix A), with most of the GO-BP terms showing enrichment in pathways related to immune and inflammatory processes, such as cell cycle, apoptotic processes, immune system development, cell proliferation, and regulation of gene expression and molecular functions.

## 3. Discussion

The present study has identified miRNAs in maternal plasma at 15 weeks’ gestation that may have utility in the early prediction of subsequent moderate to late SPTB. Importantly, we have also shown replication of the key identified miRNAs over two independent cohorts, with the top identified candidates tightly conserved over the two study groups. In addition, we have also shown an apparent global downregulation of miRNAs in the SPTB group, which is paralleled by our gene expression data showing significant downregulation of the key genes involved in miRNA biogenesis.

A novel finding in the present study is the consistent pattern of downregulation of miRNAs in the plasma of the SPTB group, which is paralleled by significant downregulation of key miRNA biogenesis genes. Global miRNA downregulation has been widely reported in the setting of some cancers [24,25,26], with some reports also linking oestrogen exposure to widespread repression of miRNA expression [27,28]. However, in our understanding, this is the first report and replication of such findings in an SPTB cohort. Interestingly, recent work by Illarionov et al. showed a predominant downregulation in plasma miRNAs, with 80% of the identified miRNAs decreased in women at high risk of PTB relative to term controls although the cohort size was small [18]. Of note, of the top differentially expressed miRNAs in the current study, there was only one overlap (miR-223-3p) with our previously published pilot data in maternal plasma at 20 weeks gestation [23], although the general pattern of miRNA downregulation in the SPTB group remains consistent. This difference is likely due to known temporal changes in miRNAs across pregnancy [19], as highlighted in the recent work by Illarionov et al. where maternal plasma miRNA profiles were distinct between first- and second-trimester timepoints [18].

Work by Cook et al. reported on the predictive value of circulating miRNAs measured at time points between 12 and 22 weeks’ gestation for subsequent preterm delivery and cervical shortening with some overlap in identified targets with the present study (e.g., miR-23a-3p, -15b-5p, -191-5p, and let-7a-5p), although directionality of change was not always consistent across studies [17]. In agreement with our observation, a recent systematic review by Subramanian et al. on miRNAs during the first trimester and later pregnancy complication also highlighted miR-191-5p as being associated with PTB [9]. Of note, the cohort demographics in Cook et al. are quite different to our study, particularly the ethnic composition (50% Caucasian in Cook et al. versus 100% Caucasian in the present study). These differences are consistent with recent reports suggesting that ethnicity contributes to inconsistent associations between miRNAs and clinical outcomes [20]. A limitation of other studies is that the miRNA profiles have been based on analysis at the time when the pregnancy complication was already manifest and thus have limited utility as early biomarkers [15]. 

Inflammation and infection are the most common factors involved in the aetiology of PTB [29]. The top candidate miRNAs identified in the current study link to key pathways involved in oxidative stress, inflammation, cell cycle, cell survival and proliferation, growth and embryonic development, angiogenesis, apoptosis, and signal transduction. Many of the top identified miRNAs also link into the serine–threonine metabolic pathways, which have recently been reported to be dysregulated in placentas in the setting of PTB [30].

miR-451a has known roles in inflammatory processes and chemotaxis [31], plays a key role in the differentiation and oxidative stress of erythrocytes [32], and is involved in processes that attenuate AKT/mTOR pathway activation [33]. In line with our findings of high plasma abundance, exosomal miR-451a has been reported to be highly expressed during pregnancy and plays a key role in the oestrogen signalling pathway [33]. Circulating miR-451a has also been reported to have some utility as part of an integrated model developed for the early prediction of pre-eclampsia [34]. 

miR-223 is proposed to play a role in inflammatory processes via repression of the NLRP (Nucleotide-binding oligomerisation domain, Leucine-rich Repeat and Pyrin domain containing Proteins)-3 inflammasome [35,36]. miR-223-3p has been suggested to play a key role in the regulation of inflammatory processes and macrophage activation via targeting of NLRP3 and PBX/Knotted 1 Homeobox 1 (Pknox1). miR-223-3p can also mediate inflammatory cytokine interleukin 1β production and anti-inflammatory responsiveness [37,38]. In particular, miR-223-3p has been proposed to be a myeloid-specific miRNA affecting immune response, haematopoiesis, and inflammatory processes and is abnormally expressed across a range of disease states [39]. Exosomal miR-223-3p in plasma has also been shown to be highly expressed during pregnancy [33] and has previously been reported to be altered in the setting of pregnancy complications, including gestational diabetes [40]. Tang et al. reported a downregulation of miR-223-3p in myometrial tissue in the setting of preterm labour but this could simply reflect a part of the normal labour process and not necessarily a precursor to PTB [41]. miR-223-3p has also been shown to play a role in uterine receptivity and diminished embryo implantation in mice via downregulation of leukaemia inhibitory factor (LIF) [42]. Cook et al. also reported that myometrial expression of mir-223-3p is regulated by oxytocin, which is a critical factor in promoting myometrial contraction [43]. Of direct relevance to the current findings, miR-223 has recently been associated with an elevated risk for PTB delivery by Winger et al. [44]. miR-223 was also shown to be associated with natural killer cell inhibition and maternal immune tolerance to pregnancy at the implantation site [45].

The Lethal-7 (Let-7) miRNAs represent one of the most widely studied miRNA families. The Let-7 miRNAs have well-established roles in cell invasion, differentiation, proliferation, migration, and metabolism [46]. Let-7a-5p has a well-recognised role in cancer, with downregulation observed across a number of cancers, but its role in pregnancy-related complications is less defined. Maternal plasma let-7a-5p has previously been shown to be altered in cervical shortening and/or PTB [17]. In a study of spontaneous miscarriages, expression of let-7a-5p was significantly reduced in decidual samples compared to those from women with uncomplicated pregnancies [47].

miR-126 has previously been shown experimentally to be highly enriched in endothelial cells and endothelial apoptotic bodies with a key role in the maintenance of vascular integrity and angiogenesis [48]. miR-126 also exerts antiapoptotic and anti-inflammatory properties [49]. Of note, miR-126-3p downregulation in maternal vascular tissue has been shown to contribute to an increased endothelial response in the setting of pre-eclampsia [50]. In the mouse, miR126-3p in epithelial cells has been reported to regulate progesterone receptors and mammary gland development [51]. 

Patterns of global downregulation in miRNAs have been reported for disease states such as some cancers [26] but this is the first report to our knowledge of such a phenomenon in the setting of SPTB. The pattern of downregulation in plasma miRNAs in the SPTB group was paralleled by a significant decrease in the expression of key genes related to miRNA biogenesis (*DROSHA*, *DICER*, and *AGO1-2*). However, there are limited data around these genes in the setting of SPTB. Of note, work by Montenegro et al. reported that the expression of DICER was markedly decreased at term, particularly with labour in the chorioamniotic membranes [52]. Their work suggested that chorioamniotic membrane expression of DICER differentiated pathological preterm labour and physiological spontaneous labour at term. The pattern of downregulation in these key miRNA regulators may also offer potential value as prognostic markers in the setting of SPTB, as has been suggested for other disorders.

### Strengths and Limitations

A key strength of this work is the sample size and replication of the observed results across two independent cohorts. Further, we only investigated SPTB, whereas prior studies have encompassed all PTB (i.e., SPTB and iatrogenic PTB) with different PTB aetiologies potentially confounding the reported outcomes. Further, the current cohorts comprised Caucasian women so have minimised the potential impacts of ethnicity on miRNA profiles as reported by others [20,21]. Of note, the average age of women in the two cohorts was significantly different (Auckland 32.6 years versus Adelaide 23.8 years) as well as women from the Adelaide cohort having an overall lower socioeconomic index score. However, the results observed were independent of age- and socioeconomic-related factors. Data reproducibility was aided by the use of standardised collection protocols and tight sampling timeframe utilised in both SCOPE study locations. Further, from the technology aspect, the majority of analyses to date have used microarray and qPCR-based approaches, which can lack the sensitivity of digital count platforms such as NanoString, as used in the present study [53]. A further strength of the work is that the gene expression data for the key biogenesis markers strongly supported the observation of decreased miRNA abundance in those mothers who later experienced SPTB. A limitation is the use of a single sampling point during pregnancy and further investigation of potential temporal changes in miRNA expression pathways across pregnancy would be informative.

In summary, the current study has identified miRNAs that may provide utility in the early prediction of SPTB and, importantly, have been replicated across two independent cohorts. Moreover, we have provided evidence for dysregulated miRNA biogenesis in early pregnancy in those women who subsequently delivered prematurely, with reduced plasma miRNA abundance concomitant with significant reductions in the expression of key genes involved in miRNA biogenesis. If these biomarkers can be independently verified across wider SPTB cohorts, those identified at risk may be able to be managed differently, including consideration of referral to specialised services, consideration of cervical cerclage or vaginal progesterone, and timely/earlier administration of adjunct therapies, including antenatal corticosteroids and magnesium sulphate, which can improve outcomes for preterm infants. Further, if the observed pattern of miRNA downregulation in the setting of SPTB holds true, this may inform potential strategies around miRNA therapeutics to mitigate deregulation of key miRNAs that may be involved in the control of pregnancy.

## 4. Materials and Methods

### 4.1. Cohorts

This study utilised participants from the Auckland and Adelaide cohorts that formed part of the Screening for Pregnancy Endpoints (SCOPE) study (Australia New Zealand Clinical Trial Registration Number ACTRN 12607000551493). SCOPE is a multi-centre prospective cohort study in healthy nulliparous women, with the primary aim of developing screening tests for prediction of a range of pregnancy-related outcomes, including SPTB. Characteristics of the SCOPE cohort have been reported in detail previously [54,55]. Cases (n = 54 (Auckland n = 36, Adelaide n = 18)) were participants who had an SPTB resulting in delivery between 28 and 36 weeks’ gestation. Matched controls (n  =  54) were women with healthy uncomplicated pregnancies with delivery ≥ 37 weeks. All participants were non-smokers and were matched by ethnicity, age  ±  3y, and BMI  ±  3 kg/m^2^. Customised birthweight centiles were adjusted for maternal height, ethnicity, and booking weight, as well as sex of the infant and gestational age at birth. Ethical approval was granted by local Ethics Committees (Auckland, AKX/02/00/364 and, in Adelaide, HREC 1712/5/2008) with written informed consent obtained from all participants. All methods were performed in accordance with the relevant ethical guidelines and regulations.

### 4.2. miRNA Analysis

Blood was collected in 6 mL sterile EDTA plasma vacutainers (Becton Dickinson (BD), New York,, USA Cat. #367873), placed on ice, followed by centrifugation (2400 g/10 minutes/4 °C). The plasma supernatant was transferred into sterile tubes as 0.25 mL aliquots and stored at −80 °C within four hours of collection. Identical collection protocols were used across both sites. RNA was extracted using the miRNeasy^®^Serum/Plasma kit (QIAGEN, Hilden Germany). RNA purity and concentration was estimated spectrophotometrically using a NanoDrop ND-100 spectrophotometer (NanoDrop Technologies, Wilmington, DE, USA). miRNA analysis was performed by Grafton Clinical Genomics (University of Auckland). In brief, miRNAs in EDTA maternal plasma at 15 weeks gestation were quantified using the NanoString nCounter human v3 miRNA expression panel (800 human miRNA targets). This assay comprises 14 assay controls (8 negative and 6 positive), 5 mRNA housekeeping controls (ACTB, B2M, GAPDH, RPL19, and RPLP0), and 5 nonmammalian miRNA spike-in probes. nCounter miRNA Expression Assay kits were used to analyse the digital detection of the plasma miRNAs in a single reaction. Digital detection was undertaken in two parts: (i) transcripts detected by probes bound to complimentary segments of DNA which are attached to a unique string of coloured fluorophores; (ii) the number of total transcripts in the sample was counted by the number of times the fluorophore was detected with scanning performed in 600 fields of view. NanoString counts represent molecules/100 ng of plasma extract. Following hybridisation, counts were analysed by the nCounter Digital Analyzer. 

Normalisation of raw data and data analysis was undertaken using the nSolver software (http://www.nanostring.com/products/nSolver) (Version 4.0, NanoString Technologies, Inc. Seattle, WA, USA, accessed 7 June 2022) [53]. In silico analysis was undertaken to examine the biological pathways associated with the top candidate miRNAs identified [56]. Receiver operator characteristic (ROC) curves were generated using SPSS (IBM, Version 29). For maternal demographic data, differences between cases (SPTB) and controls (Term) were assessed via the Fisher exact test or Student t-test. Maternal data were analysed using SigmaPlot software (V14.0, Systat Software Inc., Palo Alto, CA, USA). Data are presented as mean ± SEM unless otherwise stated. 

### 4.3. Gene Expression Analysis

The method of RNA extraction and purification from plasma was as previously described using Qiagen RNEasy Mini kits [57]. A total of 80ng RNA was reverse transcribed into cDNA using High-Capacity cDNA Reverse Transcription Kit (Thermo Fisher, Waltham, MA, USA) as per the manufacturer’s instructions. To assess the gene expression level of *Dicer1*, *Drosha*, *Ago1*, and *Ago2*, semi-quantitative real-time PCR (qPCR) was conducted on the QuantStudio 6 Flex (Thermo Fisher). TaqMan Gene Expression Assay was performed using TaqMan Fast Advance Master Mix and TaqMan probes (FAM) according to kit instructions (Thermo Fisher). The TaqMan probes used for the qPCR are as follows: *Dicer1* (4331182 Hs00229023_m1), *Drosha* (4331182 Hs00203008_m1), *Ago1* (4331182 Hs00201864_m1), *Ago2* (4448892 Hs01085580_m1), *RPS13* (4331182 Hs01011487_g1), and *RPS29* (4331182 Hs03004310_g1). The qPCR cycling program consisted of a holding step of 50 °C for 2 min, 95 °C for 20 sec, followed by 45 cycles of 95 °C for 1 sec and 60 °C for 20 s. Housekeeping genes, ribosomal protein S13 (RPS13) and S29 (RPS29), were used as endogenous controls to normalise the expression of target genes. Results were calculated using the 2^−ΔΔCt^ method [58] to determine relative quantitative levels, which were expressed as the percentage change from the relevant controls. All experiments were conducted in duplicate, and the results are presented as mean ± SEM. Statistical significance between the Case and Control groups was determined using the Mann–Whitney Rank Sum test, with a *p*-value < 0.05 considered statistically significant. Receiver operating characteristics curves (ROCs) were generated in SPSS (IBM Corp., Version 29, Armonk, NY, USA). 

### 4.4. miRNA Pathway Analysis

An integrated online platform for network-based visual analysis of miRNAs, their targets, and functions, miRNet (2.0) (https://www.mirnet.ca) (accessed 3 July 2024), was used for predicting the miRNA–gene interactions. The miRNAs that were differentially expressed across both SCOPE cohorts were chosen as input for target prediction. The miRNA–target interaction data were collected from miRTarBase v8.0 using mirBase ID. Manual curation and validation of the predicted miRNA–gene interactions were derived from the miRNA target prediction programs of miRNet. Analysis of functional enrichment was performed using a hypergeometric test using Kyoto Encyclopedia of Genes and Genomes (KEGG) and gene ontology biological process (GO-BP) terms. For the network analysis of miRNA–gene interaction, the cut-off of betweenness degree factor ≥ 2.0 was used as a filter for network connections and for the enriched pathways and FDR ≤ 0.05 was used as a filter for pathway analysis.

## Figures and Tables

**Figure 1 ijms-25-09536-f001:**
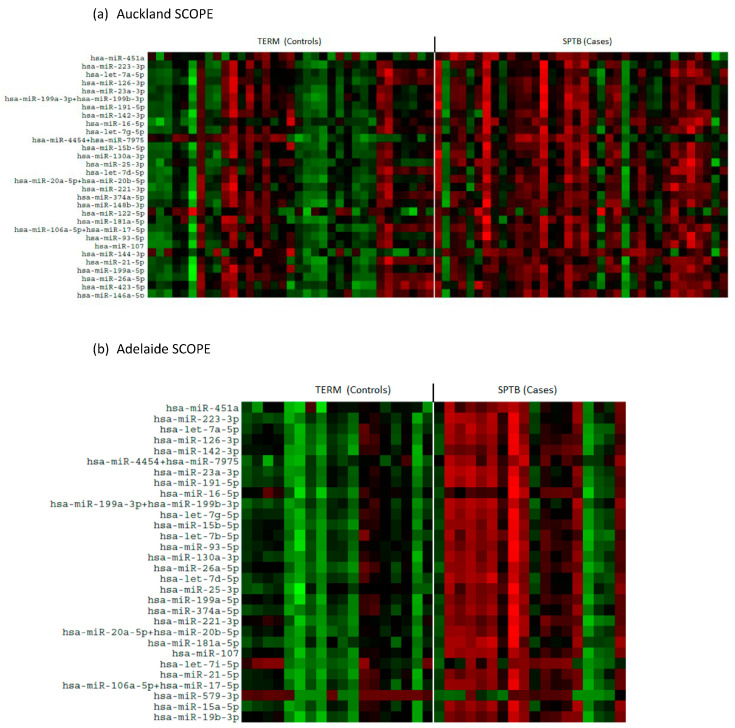
Heat maps showing differential patterns of miRNA expression between Control (Term) and Cases (SPTB). (**a**) Auckland SCOPE cohort (n = 36 per group); (**b**) Adelaide SCOPE cohort (n = 18 per group). Green = upregulated; red = downregulated.

**Figure 2 ijms-25-09536-f002:**
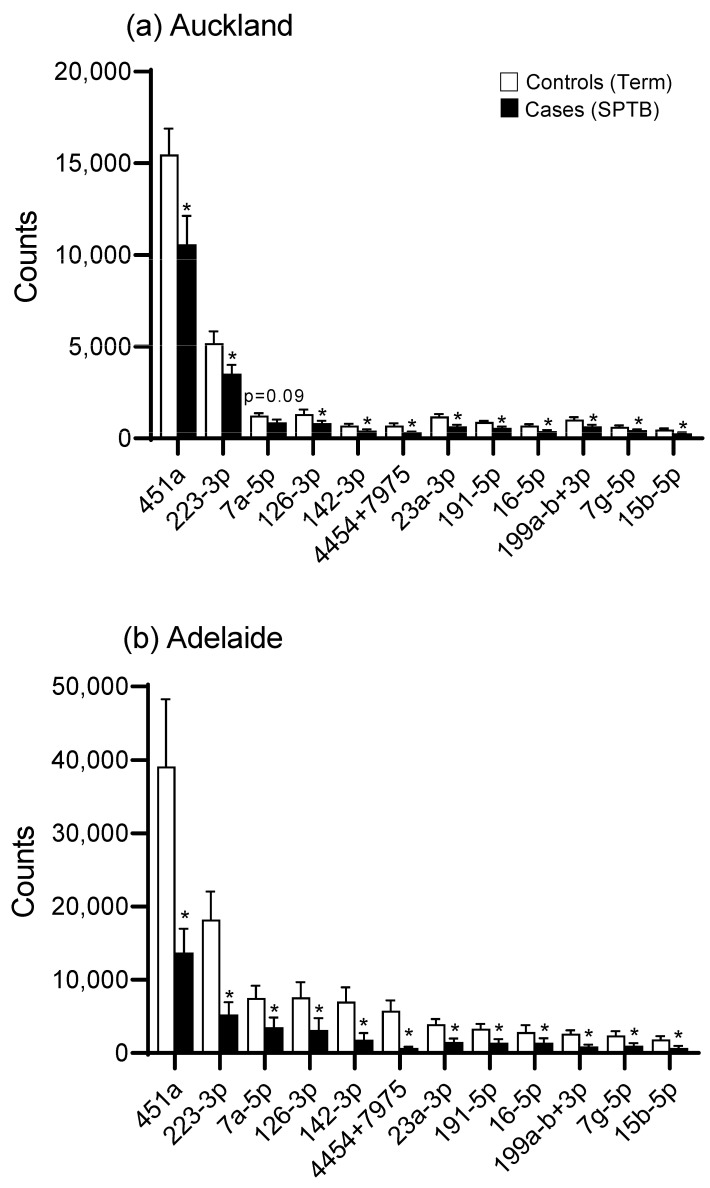
miRNA expression (expressed as digital counts) for the top 12 most highly differentially expressed miRNAs for Control (Term) versus Cases (SPTB) in (**a**) the Auckland SCOPE cohort (n = 36 per group) and (**b**) the Adelaide SCOPE cohort (n = 18 per group). Data are shown as mean ± SEM, * *p* < 0.05.

**Figure 3 ijms-25-09536-f003:**
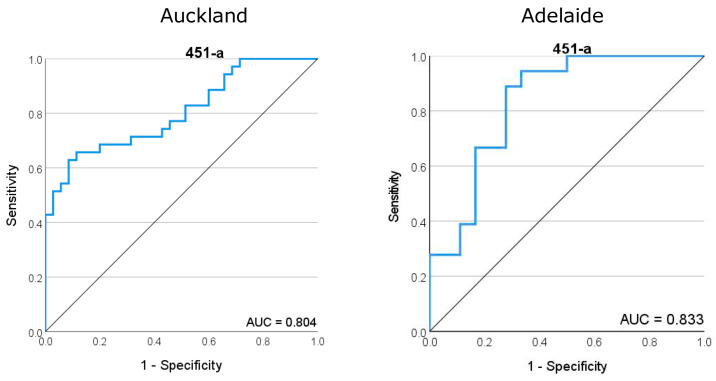
Receiver operator characteristic (ROC) curve for miR-451a, the most highly abundant miRNA in maternal plasma at 15 weeks of gestation. (**Left**) ROC analysis for the Auckland SCOPE cohort (n = 36 per group) and (**right**) ROC analysis for the Adelaide SCOPE cohort (n = 18 per group). Data for both cohorts yielded an area under the curve (AUC) score of >0.8.

**Figure 4 ijms-25-09536-f004:**
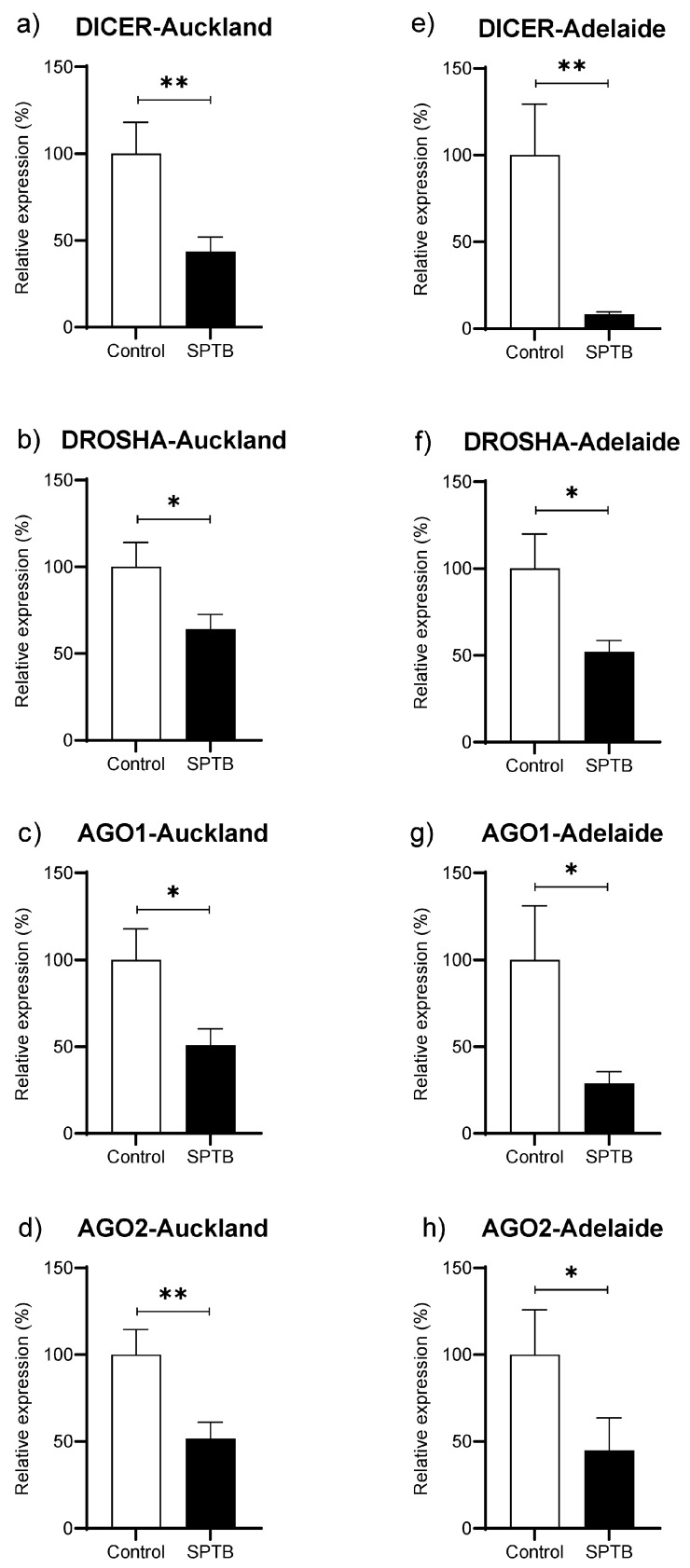
Changes in gene abundance in maternal plasma at 15 weeks of pregnancy for four key genes related to miRNA biogenesis. (**a**–**d**): gene expression of *DICER*, *DROSHA*, *Ago1*, and *Ago2* in the Auckland SCOPE cohort (n = 36 per group); (**e**–**h**): gene expression of *DICER*, *DROSHA*, *Ago1*, and *Ago2* in the Adelaide SCOPE cohort (n = 18 per group). AGO = argonaut. Data are shown as the relative change in expression in the Cases (SPTB) as compared to Controls (Term) and are presented as mean ± SEM. * *p* < 0.05, ** *p* < 0.001.

**Figure 5 ijms-25-09536-f005:**
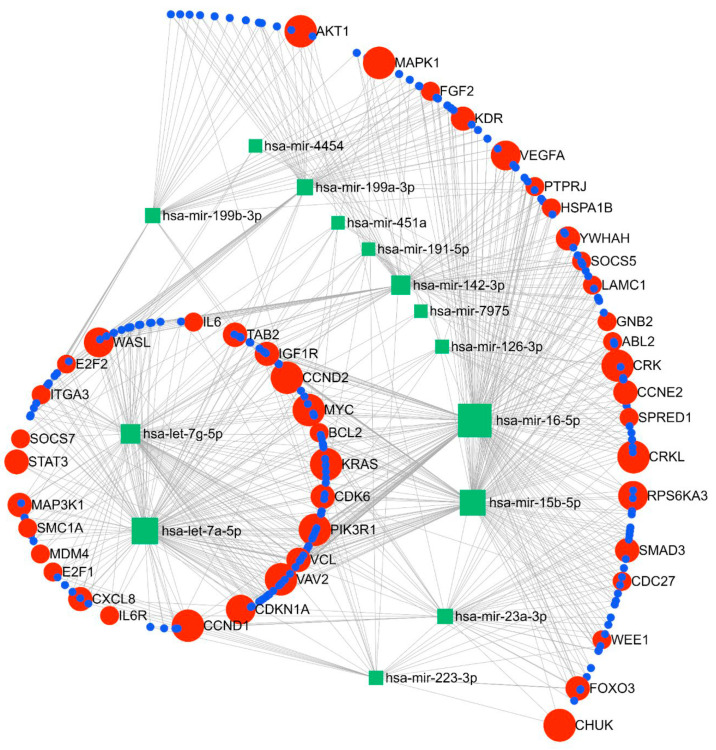
Pathway analysis showing linkage between the top 12 miRNAs and key gene targets for the identified miRNAs.

**Table 1 ijms-25-09536-t001:** Maternal demographic characteristics in the SCOPE Auckland and Adelaide cohorts. Data are presented as n (%) or mean (SEM). * Data are collected at 14–16 weeks’ gestation. *p*-values are for comparisons between the two groups using the Fisher exact test or Student *t*-test. Other than gestational age at time of delivery, there were no differences in baseline characteristics between Cases and Controls across the two maternal groups.

Auckland	Preterm n = 36	Term n = 36	*p*-Value
**Age (years) ***	32.7 (0.70)	32.4 (0.73)	0.83
**Body Mass Index (kg/m^2^) ***	24.8 (0.73)	24.6 (0.58)	0.87
**Primigravid ***	25 (69.4%)	28 (77.8%)	0.59
**Socioeconomic index score ***	52.5 (2.20)	54.2 (2.45)	0.60
**Gestation at sampling**	15.6 (0.13)	15.4 (0.13)	0.23
**Gestation at delivery (weeks)**	**33.8 (0.37)**	**40.1 (0.18)**	**<0.0001**
**Customised birthweight centile**	54.3 (4.97)	58.1 (3.60)	0.54
**Adelaide**	**Preterm** **n = 18**	**Term** **n = 18**	
**Age (years) ***	24.6 (1.39)	22.9 (1.06)	0.35
**Body Mass Index (kg/m^2^) ***	28.3 (1.56)	26.6 (1.34)	0.42
**Primigravid ***	12 (66.7%)	16 (88.9%)	0.23
**Socioeconomic index score ***	33.4 (2.90)	31.5 (3.06)	0.66
**Gestation at sampling**	15.6 (0.14)	15.7 (0.12)	0.97
**Gestation at delivery (weeks)**	**33.8 (0.82)**	**40.5 (0.19)**	**<0.0001**
**Customised birthweight centile**	52.2 (7.07)	53.7 (6.27)	0.88

**Table 2 ijms-25-09536-t002:** The top 12 most abundant miRNAs in 15-week maternal plasma in the Auckland and Adelaide SCOPE cohorts. The most highly expressed miRNAs are highly conserved across both cohorts.

Rank	Auckland	Adelaide
1	hsa-miR-451-a	hsa-miR-451-a
2	hsa-miR-223-3p	hsa-miR-223-3p
3	hsa-let-7a-5p	hsa-let-7a-5p
4	hsa-let-126-3p	hsa-let-126-3p
5	hsa-miR-23a-3p	hsa-miR-142-3p
6	hsa-miR-199a-3p-hsa-miR-199b-3p	hsa-miR-4454-hsa-miR-7975
7	hsa-miR-191-5p	hsa-miR-23a-3p
8	hsa-miR-142-3p	hsa-miR-191-5p
9	hsa-miR-16-5p	hsa-miR-16-5p
10	hsa-miR-7g-5p	hsa-miR-199a-3p-hsa-miR-199b-3p
11	hsa-miR-4454-hsa-miR-7975	hsa-miR-7g-5p
12	hsa-miR-15b-5p	hsa-miR-15b-5p

**Table 3 ijms-25-09536-t003:** Gene pathways and pathway target genes as identified via miRNA–gene network and functional analysis using miRNet Version 2.0 (https://www.mirnet.ca) accessed on 3 July 2024.

Gene Pathway	Pathway Target Genes
Cell Cycle	CCND1, CCND2, CDK6, CDKN1A, E2F1, E2F2, MYC, SMC1A, CDC27, SMAD3, WEE1, YWHAH, CCNE2
Focal Adhesion	CCND1, BCL2, CCND2, IGF1R, ITGA3, PIK3R1, VAV2, VCL, CRK, CRKL, KDR, LAMC1, VEGFA, AKT1, MAPK1
Jak-STAT signaling	CCND1, CCND2, IL6R, MYC, PIK3R1, STAT3, SOCS7, SOCS5, SPRED1, AKT1
ErbB signalling	CDKN1A, KRAS, MYC, PIK3R1, ABL2, CRK, CRKL, AKT1, MAPK1
Chemokine signaling	CXCL8, KRAS, PIK3R1, STAT3, VAV2, WASL, CHUK, CRK, CRKL, GNB2, FOXO3, AKT1, MAPK1
Neurotropin signaling	KRAS, MAP3K1, PIK3R1, CRK, CRKL, RPS6KA3, YWHAH, FOXO3, AKT1, MAPK1
mTor signaling	PIK3R1, RPS6KA3, VEGFA, AKT1, MAPK1
P53 signaling	CCND1, CCND2, CDK6, CDKN1A, MDM4, CCNE2
Adherens junction	IGF1R, VCL, WASL, SMAD3, PTPRJ, MAPK1
Fc gamma R-mediated phagocytosis	PIK3R1, VAV2, WASL, CRK, CRKL, AKT1, MAPK1
B cell receptor signaling	KRAS, PIK3R1, VAV2, CHUK, AKT1, MAPK1
VEGF signaling	KRAS, PIK3R1, KDR, VEGFA, AKT1, MAPK1
MAPK signaling	KRAS, MAP3K1, MYC, TAB2, CHUK, CRK, CRKL, FGF2, HSPA1B, RPS6KA3, AKT1, MAPK1
Toll-like receptor signaling	IL6, CXCL8, PIK3R1, TAB2, CHUK, MAPK1
T cell receptor signaling	KRAS, PIK3R1, VAV2, CHUK, AKT1, MAPK1
Fc epsilon RI signaling	KRAS, PIK3R1, VAV2, AKT1, MAPK1

## Data Availability

Data are available upon reasonable request.

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
