# Peer review of "Maternal Plasma miRNAs as Early Biomarkers of Moderate-to-Late-Preterm Birth"

_ijms, 2024, doi:10.3390/ijms25179536_

Round 1

Reviewer 1 Report

Comments and Suggestions for Authors

This is an interesting work on the use of miRNAs as predictive plasma biomarkers of preterm birth. The authors have tested two different cohorts of subjects and the results obtained substantiate a global deregulation of miRNA maturation and expression. Nevertheless, this work has more of a pilot project than of a finished one: the experimental branch is very poor, just one determination of miRNA levels, while the “in silico” part is meager and insufficient to draw any conclusion.  

Major points.

-Although the experimental design is well done, I have a general concern regarding the miRNA detection technology that seems to be under-detecting plasma miRNAs, since the first test (Figure 1) only reports expression of over 30 miRNAs and only 12 of them (out of 800!!!) were confirmed upon re-testing, from which I would trust only two (miR-451a and miR-223-3p). Could be this due to the low amount of miRNAs in plasma? to sample degradation during preparation?. The authors should clarify this point (did they use any control of integrity?).

- There is a lack of coherency among the expression and “in silico” analysis. While in the former only 12 miRNAs are studied in the later there appear 14 miRNAs since 4454 and 7975 are treated separately. Is there any reason for that?

-- The Discussion section is almost a “miRNA catalogue” description and should be re-written. I would center in miR-451a and miR-223 that are the most promising ones.

- The “in silico” part is very poor. Just a KEGG analysis and a list of putative target mRNAs. Again, I would suggest the authors to center in miRs 451 and 223, making a deeper analysis (e.g. in the  mirSystem server) of pathways and/or putative target mRNAs.

Minor points

-Commercial brands should be correctly referred: (Brand name, City, State (if any), country) Furthermore, in the M&M section these are referred in different variants, these should be normalized.

- Figure 5 is non informative (none of the target genes is identified) and can be deleted.

-I can’t see the interest of including the results of Drosha/Dicer/Ago expression. Only two lines in the Results section and none in the Discussion section. Furtermore, I can’t see the relationship among plasma levels of miRNAs and of D/D/A. This should be discussed.

Reviewer 2 Report

Comments and Suggestions for Authors

Thank you for the opportunity to review the study.

The study was well-organized and understandable.

I have several requests to the authors.

  1. As there are several pathologies in SPTB, if there are any information about pathological findings of placenta (inflammation), please describe in the paper.
  2. The study includes two cohorts. The authors write about demographic differences in miRNA profile, therefore, please add demographic information.
  3. Please change “Fisher`s Exact Test” , better to write as “Fisher exact test”?

Author Response

Thank you for the opportunity to review the study.

The study was well-organized and understandable.

I have several requests to the authors.

  1. As there are several pathologies in SPTB, if there are any information about pathological findings of placenta (inflammation), please describe in the paper.
  2. The study includes two cohorts. The authors write about demographic differences in miRNA profile, therefore, please add demographic information.
  3. Please change “Fisher`s Exact Test” , better to write as “Fisher exact test”?

Reviewer 3 Report

Comments and Suggestions for Authors

The authors analyze maternal plasma miRNAs as early biomarkers of moderate-to-late preterm birth

This scientific article is very interesting because it evaluates some factors of epigenetic modulation important in preterm birth. Interestingly during this evaluation the authors indirectly demonstrate that the variation of these factors is not dependent on ethnicity.

The authors rightly point out that two types of preterm births can be classified: spontaneous (SPTB) or iatrogenic PTB but in reality they do not describe in the introduction what is meant by spontaneous preterm birth and what can be defined as iatrogenic. Among iatrogenic preterm births the authors should discriminate between two subclasses, namely those induced but unwanted after amniocentesis and chorionic villus sampling and those induced and wanted due to fetal growth restriction.

It is very important to recognize the miRNA pathway in the pathogenesis of preterm birth, but the authors should better explain how to combine the fact that preterm birth involves inflammatory processes, and at the same time the authors demonstrate that miRNAs involved in inflammatory pathways are downregulated during preterm birth.

And furthermore, if the inflammatory process is fundamental in preterm birth, what quantitative relationship is there for these downregulated miRNAs with sudden intrauterine fetal death (SIUD)?

Authors may enrich the references by adding information on the definition and etiopathogenesis of preterm birth, on the implication of miRNAs, if any, with SIUD.

The tables are appropriate

Reviewer 4 Report

Comments and Suggestions for Authors

The presented work is highly interesting and addresses a topic of significant clinical and scientific relevance. The research contributes substantially to the understanding of preterm birth risk, providing valuable data and thorough analyses. However, there are a few suggestions that could further enhance the manuscript.

1. Introduction: In the introduction, I suggest including a recent and highly relevant scientific finding: the risk of preterm birth is increased regardless of the mode of conception. Please add this reference: doi: 10.1016/j.ajog.2024.05.037. This information not only enriches the context of the study but also helps to bypass the limitation of potential selection bias, as the mode of conception for the selected cohort is not specified in the analysis.

2. Materials and Methods

The description of the study cohort and controls is detailed and well-structured. However, the information about the cases and matched controls, including the demographic details, should be moved to the results section and presented along with a comparison table between the two study cohorts (SPTB vs. term birth).

3. Results

The results are clear and well-presented. However, I suggest expanding this section to include detailed characteristics of the study cohort. This will enrich the analysis and provide a stronger basis for the conclusions drawn.

4. Discussion

The discussion is good and comprehensive, addressing the key points of the study and highlighting the clinical and scientific implications of the findings. Excellent work on this section.

5. Figures

The quality of Figure 2 deserves revision. Ensuring it is of high quality and that the data is presented clearly and comprehensibly will help strengthen the visual impact and understandability of the work.

Conclusion

With the suggested modifications, I believe the work can be accepted for publication. Congratulations on the excellent work on such a highly relevant topic

Round 2

Reviewer 1 Report

Comments and Suggestions for Authors

Authors have make an effort to improve the manuscript.